# Lifestyle Changes and Body Mass Index during COVID-19 Pandemic Lockdown: An Italian Online-Survey

**DOI:** 10.3390/nu13041117

**Published:** 2021-03-29

**Authors:** Silvia Maffoni, Silvia Brazzo, Rachele De Giuseppe, Ginevra Biino, Ilaria Vietti, Cristina Pallavicini, Hellas Cena

**Affiliations:** 1Clinical Nutrition and Dietetics Service, Unit of Internal Medicine and Endocrinology, ICS Maugeri IRCCS, 27100 Pavia, Italy; silvia.maffoni@icsmaugeri.it (S.M.); silvia.brazzo@icsmaugeri.it (S.B.); ilaria.vietti@icsmaugeri.it (I.V.); cristina.pallavicini@icsmaugeri.it (C.P.); 2Laboratory of Dietetics and Clinical Nutrition, Department of Public Health, Experimental and Forensic Medicine, University of Pavia, 27100 Pavia, Italy; rachele.degiuseppe@unipv.it; 3Institute of Molecular Genetics, National Research Council of Italy, 27100 Pavia, Italy; ginevra.biino@igm.cnr.it

**Keywords:** lifestyle, body mass index, COVID-19 pandemic

## Abstract

Background. COVID-19 pandemic has imposed a period of contingency measures, including total or partial lockdowns all over the world leading to several changes in lifestyle/eating behaviours. This retrospective cohort study aimed at investigating Italian adult population lifestyle changes during COVID-19 pandemic “Phase 1” lockdown (8 March–4 May 2020) and discriminate between positive and negative changes and BMI (body mass index) variations (ΔBMI). Methods. A multiple-choice web-form survey was used to collect retrospective data regarding lifestyle/eating behaviours during “Phase 1” in the Italian adult population. According to changes in lifestyle/eating behaviours, the sample was divided into three classes of changes: “negative change”, “no change”, “positive change”. For each class, correlations with ΔBMI were investigated. Results. Data were collected from 1304 subjects (973F/331M). Mean ΔBMI differed significantly (*p* < 0.001) between classes, and was significantly related to water intake, alcohol consumption, physical activity, frequency of “craving or snacking between meals”, dessert/sweets consumption at lunch. Conclusions. During “Phase 1”, many people faced several negative changes in lifestyle/eating behaviours with potential negative impact on health. These findings highlight that pandemic exacerbates nutritional issues and most efforts need to be done to provide nutrition counselling and public health services to support general population needs.

## 1. Introduction

COVID-19 pandemic has imposed a period of contingency measures, including total or partial lockdowns all over the world, with different modalities according to the countries. In Italy the first lockdown, “Phase 1” (DPCM-GU Serie Generale n.59; 8 March 2020), occurred March 8. This phase has greatly impacted everyday life of all citizens, leading to different adaptive strategies.

Emerging studies [1,2,3,4] have assessed the impact of this situation on physical and mental health. Di Renzo et al. [1] analyzing the effect of COVID-19 pandemic on eating habits and lifestyle in a sample of Italian respondents (age range, 12 and 86 years), by means of an online survey, reported that young adults (aged 18 to 30) adhered to the Mediterranean dietary pattern more than youth and elderly, with 15% of the respondents turning to organic fruit and vegetables [1]. Furthermore, other virtuous behaviors were reported by the authors as a slight increase in physical activity as well as a reduction in smoking habits [1]. Others [2] instead have suggested that social isolation has led to consuming more ultra-processed, energy-dense comfort foods, purchasing more packaged and shelf-stable foods, both to reduce trips outside home to the supermarket and to have a richer and more satisfying pantry. At the same time, social distancing and home deliveries decreased the opportunities for physical activity, especially in individuals living in small apartments in urban areas, increasing sedentariness, and affecting health and wellbeing [2]. However, it is important to emphasize that numerous interindividual differences account for different behaviors [3] that may depend on personal characteristics and that have led some to cook more and devote more time to meals within the family and others instead to use the food in a disorderly way for boredom, as a consolation or to quell anxiety [3].

This pandemic has more than ever mastered the notion that there are inevitable repercussions of the general malaise on mental wellbeing. A large multicenter Italian study [4] explored the implications of social isolation on psychological distress in the academic population of five Universities, reporting that about 20% of the participants showed severe levels of anxiety and mood deflection. 

Therefore, based on these previous observations [1,2,3,4], the primary aim of our study was to investigate lifestyle habits and eating behaviors modifications in a sample of Italian adults during “Phase 1” COVID-19 pandemic home confinement. The secondary aim was to discriminate between positive and negative changes in lifestyle habits and eating behaviors and their relationship with body mass index (BMI) variations, raising awareness of the need for public health actions to elicit positive behavior changes and prevent negative behavior changes in individuals.

## 2. Materials and Methods

### 2.1. Online-Survey

A 38 multiple-choice web-form survey in Google Forms was used to collect retrospective demographic and anthropometric data as well as lifestyle habits and eating behaviors during the first lockdown phase of COVID-19 pandemic (“Phase 1”, 8 March–4 May 2020) in Italy. The survey was launched on 30 April 2020, through WhatsApp, institutional social networks channels (e.g., Facebook and LinkedIn) and our Lab (Laboratory of Dietetics and Clinical Nutrition) mailing list. Adults (>18 years) residing in Italy were eligible to participate. Data collection ended on 10 May 2020.

This web-form administration did not allow a probability sampling procedure; however, it was effective for the research objectives since it facilitated wide dissemination of the survey.

The web-survey was conducted in agreement with the national and international regulations and the Declaration of Helsinki. All participants were fully informed about the study objectives and requirements and were asked to provide informed consent accepting the data sharing and privacy policy, before accessing the survey. Participants completed the survey by connecting directly to Google Forms, a web-based app used to create forms for data collection purposes and which is included as part of the free Google Docs Editors suite offered by Google [5]. All the surveys were then downloaded as Microsoft Excel sheet, with the maximum guarantee of anonymity inherent in the web-survey format which does not allow to trace sensitive personal data in any way [6]. For this reason, this study is exempt from the request of the ethics committee approval. 

### 2.2. Variables Collected

To investigate changes experienced during “Phase 1” COVID-19 pandemic home confinement, participants were asked to describe their lifestyle habits and eating behaviors, before “Phase 1” (8 March 2020, T_0_) and during “Phase 1” (8 March–4 May 2020, T_1_). For this purpose, in the present research, we considered 10 multiple-choice items, out of the 38 ones, exploring (i) physical activity; (ii) daily water consumption; (iii) daily caffeine consumption; (iv) daily alcohol consumption; (v) daily breakfast consumption; (vi) habitual sandwich/pizza consumption at lunch; (vii) habitual sweets/dessert consumption at lunch; (viii) habitual fruit consumption at lunch; (ix) habitual vegetable consumption at lunch; (x) “craving or eating between meals” habit. 

The survey also collected demographic information (such as gender and residency) as well as anthropometric data (such as weight and height) useful to calculate BMI (Kg/m^2^) both before and during COVID-19 pandemic “Phase 1”. 

### 2.3. Scoring

Each question investigating lifestyle habits and eating behaviors included multiple choices coded from a score ranging from 0 to 2, where 0 = negative change (shifting away from the national dietary guidelines) [7]; 1 = no change; 2 = positive change (shifting towards the national dietary guidelines [7]. The maximum achievable final score was 20.

The total score was then divided into tertiles. According to the tertiles, subjects were classified into three different “classes of change” that occurred during “Phase 1”: (i) subjects with negative change, (shifting away from the national dietary guidelines) [7] (subjects scoring < 10; 1st tertile); (ii) subjects with no change (subjects scoring 10 or 11; 2nd tertile); and finally (iii) subjects with positive change (shifting towards the national dietary guidelines) [7] (subjects scoring ≥ 12 and ≤16; 3rd tertile).

### 2.4. Statistical Analysis

Basic description of data and statistical analyses were performed using STATA 16.1 (Stata Corp LLC, College Station, TX, USA). BMI mean values at T_0_ and T_1_, between North and Centre-South regions and, between genders were compared by *t*-test. Multiple regression analysis was used to evaluate what lifestyle factors (independent variables simultaneously put in the model) were significantly related to BMI change (dependent variable), adjusting for sex. 

## 3. Results

We received 1360 questionnaires and selected 1304 subjects (973F/331M) after removing potential duplicates by comparing their IDs. Most of the sample (82.7%, *n* = 1078) resided in the northern regions of Italy while 17.3% (*n* = 226) in the Central and Southern ones. 

We observed missing item responses only in item #9 (“Habitual vegetable consumption at lunch”), with a total of 1241/1304 responses (4.8% missingness).

As reported in Table 1, BMI increased significantly at T_1_ (*p* < 0.0001). The median value of the lifestyle habits and eating behaviors total score showed that most of the subjects belonged to the 2nd tertile, meaning substantially that no changes occurred during “Phase 1”; no significant differences were recorded between residents in different regions (Northern vs. Centre + Southern) nor between gender (Table 1). Changes in lifestyle habits and eating behaviors during “Phase 1” are also reported in Table 1.

Multiple regression analysis (Table 2) was conducted to evaluate which lifestyle and eating behavior variables were significantly related to BMI variations (ΔBMI). ΔBMI (Kg/m^2^) as dependent variable and lifestyle habits and eating behaviors (e.g., physical activity; adequate daily water consumption; alcohol consumption; caffeine consumption; daily breakfast consumption; habitual sandwich/pizza consumption at lunch; habitual sweets/dessert consumption at lunch; habitual fruit consumption at lunch; habitual vegetable consumption at lunch and craving or eating between meals) as independent ones were simultaneously assessed. The analysis showed that BMI increase (ΔBMI > 0) during “Phase 1” was significantly and negatively related to the following behaviors: (i) inadequate water consumption (β: −0.09; SE: 0.04; *p* = 0.01); (ii) excessive alcohol consumption (β: −0.2; SE: 0.04; *p* < 0.000); (iii) decreased physical activity (β: −0.12; SE: 0.03; *p* = 0.000); (iv) increased frequency of “craving or eating between meals” (β: −0.28; SE: 0.04; *p* < 0.000); (v) habitual consumption of dessert/sweets at lunch (β: −0.37; SE: 0.07; *p* < 0.000).

In the three classes of change, mean (± SD) BMI value did not differ significantly (“negative change”: 23.34 ± 4.19; “no change” 23.23 ± 4.15; “positive change” 23.09 ± 4.01, *p* = 0.737). On the contrary, mean (± SD) ΔBMI differed significantly (*p* < 0.001) between subjects in the “negative change” class (0.4 ± 10.8), subjects in the “no change” class (0.12 ± 0.74), and subjects in the “positive change” class (−0.18 ± 0.9), as reported in Figure 1.

## 4. Discussion

Home confinement experienced during COVID-19 pandemic has undeniably changed everyday life of most people. Some of these changes have affected lifestyle, known as the set of individual habits and behaviors, including dietary pattern and physical activity [1,2,3,4] with consequences on body weight among the many [1,3]. The present brief report illustrates lifestyle habits and eating behaviors modifications in a sample of Italian adults during COVID-19 pandemic “Phase 1” home confinement, discriminating between positive and negative changes, according to National dietary guidelines and their association with ΔBMI.

Our sample population was characterized overall by a slight but significant BMI increase during COVID-19 pandemic “Phase 1”. After classifying lifestyle changes into three classes (negative changes, no changes, positive changes), we observed a significant variation of BMI (ΔBMI) between the three groups (*p* < 0.001). The groups experiencing negative lifestyle changes and the one experiencing positive ones reported an increased and decreased BMI respectively, while the one experiencing no lifestyle changes reported no substantial BMI modifications. In particular, those subjects who reported negative changes of their behaviors, namely “adequate daily water consumption”, “alcohol consumption”, “physical activity”, “craving or eating between meals”, “habitual sweets/dessert consumption at lunch”, were also those who displayed a significant increase in BMI during “Phase 1”.

Considering the whole sample, it is important to highlight that around one-third of the examined population worsened their lifestyle behaviors.

These results suggest that although most people show good resilience, still a large part of the community is at risk and predisposed to acquire lifestyle behaviors that are potentially harmful to their health, regardless of the initial body weight. Indeed, we observed that while some took advantage of the home confinement to increase physical activity, many people stopped or reduced sports and recreational activities also facilitated by the closure of gyms and swimming pools. These findings suggest the need to raise awareness of subjects regarding lifestyle according to national guidelines [8], and to tailor strategies for health policies.

Comfort food consumption, including desserts or sweets at lunch, is positively associated with BMI increase, suggesting that lockdown induced people to spend more time planning meals, cooking tasty foods and homemade sweets, as already reported by other surveys [1]. A positive association between BMI and “craving or eating between meals” as well as consumption of unhealthy snacks and drinks for stress, anxiety and time spent in front of a TV screen was observed, consistently with what reported by others [9,10].

In a survey conducted in Spain during home confinement, where the sample was grouped in three clusters based on food and cooking habits (i. *self-control*, with restraint, attitude on food and cooking habits; ii *sensitive*, with emotional attitudes on food and cooking habits; iii *non-emotional*, with no emotional attitudes on food and cooking habits), the authors reported that the “*sensitive*” one, ate more and often and snacked more between meals, influenced by a lower mood due to social isolation.

Noteworthy snacking and increased ultra-processed foods consumption in association with mood disorders may be confused with emotional eating, not allowing a subclinical eating disorder, implicated in the circuit of food reward and addiction to be correctly diagnosed [11].

Finally, it is well known that stressful events, including home confinement, may also induce people to greater consumption of alcoholic beverages [12], which promotes a positive energy balance and contributes to weight gain [13]. In a survey conducted in Poland during lockdown, alcohol consumption was increased in 14.6% of the sample, with higher rates of alcohol abuse in adults [12]. Our results showed that 16.4% of the sample, increased their habitual alcohol consumption, especially those who already abused it. Alcohol consumption does not protect in any way from infections, including SARS CoV2, indeed it impairs immune response with a dose-dependent correlation [14]. Isolation and drinking can also increase the risk of self-injury, suicide and violence, especially domestic violence against women, leading them in turn to drink more in a vicious circle [14]. That’s why it is necessary to conduct public health information campaigns to raise population awareness of all the risks related to excessive alcohol consumption during isolation and provide interventions to protect the most vulnerable ones [14].

Despite this study highlights the need to address future health emergencies with public health actions considering also lifestyle, authors acknowledge some limitations. First of all, the data are self-reported and this leads to potential bias, moreover it is not possible to check data accuracy since participants in Google forms are anonymous and may easily provide false information. The sample was not representative of specific Regions nor of the whole country. Although, the survey was conducted on the adult population (aged > 18 years), age was not registered, and the analysis was not age-adjusted. Finally, the authors investigated lifestyle habits and eating behavior changes to simply discriminate between positive and negative changes and their relationship to BMI during COVID-19 pandemic, unable to exclude pre-existing eating or mood disorders.

## 5. Conclusions

This study showed that during “Phase 1” COVID-19 pandemic home confinement, several changes in lifestyle habits and eating behaviors occurred, with individual differences probably depending on personal resilience. However, the short duration of “Phase 1” did not allow to highlight the impact of these changes on long term outcomes Nevertheless, we underline the need to increase public health actions to meet emergency needs and reduce vulnerability over the long term through expanded social protection, health care and education, especially for the most vulnerable groups including seniors, children and women, and for those with poor access to essential goods such as food, education and health care.

This is not going to be the last pandemic we face; therefore, we need to build resiliency in our population and nutrition is a key factor. Public health interventions should consider the need to decrease levels of inequity and protect people from further health threats and disease. Further research to understand if and to what extent these changes have regressed, or have remained stable, in the long term and what impact they have had on health is recommended.

## Figures and Tables

**Figure 1 nutrients-13-01117-f001:**
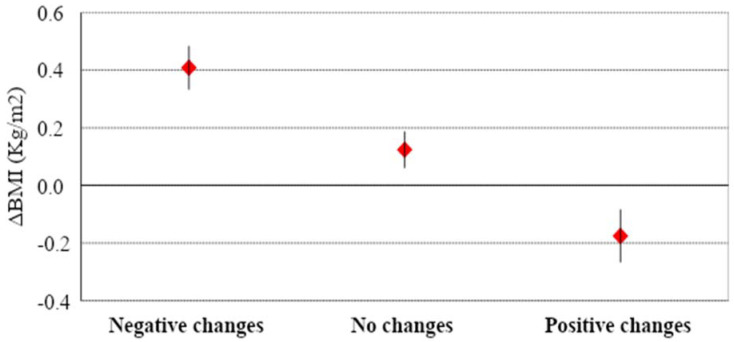
Differences in BMI (ΔBMI) concerning the 3 classes of changes (data are means along with 95% CI). *p* < 0.001: Analysis of Variance (ANOVA). Legend. Negative changes: subjects who negatively changed their lifestyle habits and eating behaviors during COVID-19 pandemic “Phase 1” (total score < 10); No changes: subjects who did not change their lifestyle habits and eating behaviors during COVID-19 pandemic Phase 1” (10 ≤ total score ≤ 11); Positive changes: subjects who positively changed their lifestyle habits and eating behaviors during COVID-19 pandemic “Phase 1” (12 ≤ total score ≤ 16).

**Table 1 nutrients-13-01117-t001:** Description of BMI, lifestyle habits and eating behaviors during COVID-19 pandemic “Phase 1”. For BMI, lifestyle habits and eating behaviors score, data are described as mean (standard deviation, SD), median (InterQuartile Range, IQR) and minimum (min.) and maximum (max.) value. For lifestyle habits and eating behaviors changes, data are described as absolute number (n) and relative frequency (%) of subjects who changed their lifestyle habits and eating behaviors.

**BMI**			
***n* = 1304 (973F/331M)**	**Mean (SD)**	**Median (IQR)**	***Min.–Max.***
BMI (Kg/m^2^) T_0_	23.2 (4.1)	22.5 (20.3–25.2)	15.8–48.4
BMI (Kg/m^2^) T_1_	23.4 (4.1)	22.7 (20.4–25.3)	15.6–50.2
*p* (T_0_ vs. T_1_)	<0.0001		
ΔBMI (Kg/m^2^)	0.15 (0.8)	0.2 (0.00–0.63)	−4.95–+5.31
**Lifestyle Habits And Eating Behaviours Score**			
**Score**	**Mean (SD)**	**Median (IQR)**	***Min.–Max.***
Total	10.22 (1.89)	10 (2)	4–16
North	10.19 (1.87)	10 (2)	4–16
Centre + South	10.4 (1.97)	10 (2)	5–16
*p (North* vs. *Centre + South)*	*0.136*		
Females	10.23 (1.9)	10 (2)	4–16
Males	10.21 (1.85)	10 (2)	5–16
*p (males* vs. *females)*	*0.883*		
**Lifestyle Habits and Eating Behaviours Changes**			
*n* = 1304 (973F/331M)	**Classes of Changes**		
	Negative Change Number (%) of subjects with negative lifestyle changes (total score < 10) *	No Change Number (%) of subjects with no lifestyle changes (total score of 10 or 11) *	Positive Change Number (%) of subjects with positive lifestyle changes (total score ≥ 12 and ≤16) *
1. Physical activity	397 (30.4)	556 (42.6)	351 (26.9)
2. Adequate daily water consumption	269 (20.6)	792 (60.7)	243 (18.6)
3. Caffeine consumption	103 (7.9)	857 (65.7)	344 (26.4)
4. Alcohol consumption	214 (16.4)	820 (62.9)	270 (20.7)
5. Daily breakfast consumption	52 (4.0)	1162 (89.1)	90 (6.9)
6. Habitual sandwich/pizza consumption at lunch	17 (1.3)	1239 (95.0)	48 (3.7)
7. Habitual sweets/dessert consumption at lunch	128 (9.8)	1155 (88.6)	21 (1.6)
8. Habitual fruit consumption at lunch	61 (4.7)	1076 (82.5)	167 (12.8)
9. Habitual vegetables consumption at lunch	94 (7.6)	1015 (81.9)	132 (10.6)
10. “Craving or eating between meals”	210 (16.1)	859 (65.9)	235 (18.0)

Significance: *p* < 0.05; t-test analysis. Legend. * Each question investigating lifestyle habits and eating behaviors included multiple choices coded from a score ranging from 0 to 2 where 0 = negative change (shifting away from the national dietary guidelines) [7]; 1 = no change; 2 = positive change (shifting towards the national dietary guidelines [7].

**Table 2 nutrients-13-01117-t002:** Multiple regression analysis: ΔBMI (Kg/m^2^) was the dependent variable while lifestyle habits and eating behaviors scores were the independent ones.

Lifestyle Habits and EATING Behaviors Scores	β (SE)	*p*-Value
Physical activity	−0.12 (0.03)	0.0001
Adequate daily water consumption	−0.09 (0.04)	0.0130
Alcohol consumption	−0.2 (0.04)	0.0000
Caffeine consumption	0.02 (0.04)	0.6370
Daily breakfast consumption	0.12 (0.07)	0.0850
Habitual sandwich/pizza consumption at lunch	0.01 (0.1)	0.9420
Habitual sweets/Desert consumption at lunch	−0.37 (0.07)	0.0001
Habitual fruit consumption at lunch	0.01 (0.05)	0.8870
Habitual vegetables consumption at lunch	−0.04 (0.05)	0.4710
Craving or eating between meals	−0.28 (0.04)	0.0000

Significance: *p* < 0.05; multiple regression analysis.

## Data Availability

All data presented in this study, not yet publicly archived, shall be made available through the corresponding author on request.

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
