# Peer review of "Lifestyle Changes and Body Mass Index during COVID-19 Pandemic Lockdown: An Italian Online-Survey"

_nutrients, 2021, doi:10.3390/nu13041117_

Round 1

Reviewer 1 Report

It is an interesting study on changes in lifestyle and anthropometric measures between before and during lockdown in Italy. The title does not reflect this and should be self-explanatory. The study relies on a self-reported retrospective questionnaire, subject to a large amount of bias.  

Language needs revision, it is inaccurate in may instances, and lacks continuity, sometimes lacks of verbs. Inappropriate use of connectors such as Nevertheless. Example of issues (but the entire manuscript needs extensive revision): 

Line 37: what is "smart working"?

Line 38: "studies are raising" --> emerging

One-sentence paragraphs should be avoided. The introduction is made up of 4 one-sentence paragraph which do not link well with each other. 

Line 43: Why is this sentence at present tense? Are they the results from a study, from which country / setting?

Intro and abstract: Some countries in the world did not apply strict lockdowns, I would talk about "contingency measures including total or partial lockdowns". 

Line 37: what is "smart working"?

Line 111: Multiple regression should be multivariable? All lifestyle factors included in the same model? Or different regression models separately?

Some information are missing on the study population. Is it intended to be representative of the population of specific region, all of Italy..? Very little control is possible to check for accuracy of age, sex, place of residence, employment, as participants in the Google forms are anonymous and can easily report false information. 

How were "potential duplicates" identified ? (line 116)

Table 1. It is hard to understand what is described here. The Negative change / no change / positive change is for each individual lifestyle habit? So that does not correspond to a total score <10, 10-11, >=12. Rather it should be individual score of 0, 1, or 2. 

The results of the "multiple regression" are not presented in any table or figure, just some of them are mentioned in the text. These data need to be presented, and the models clarified.

Also, some more details are needed on the time frame. T0 "pre-lockdown" is quite a vague date. What was the exact question? Over the month of January/February 2020? 

Line 165: what do you mean by "as previously reported"? In another study? Reference?

Line 168: "BMI was significantly negatively associated" is wrong wording. It should be something like "People who worsened some of their behaviours, namely water consumption, alcohol etc, displayed an increase in BMI compared to pre-lockdown".

Some ideas are good in the discussion but the formulation is shaky. E.g. line 200 trying new food and recipes is not a "downside". 

The sentence line 207 to 212 is very long, lacks a verb and does not make a lot of sense.

Why are some results discussed in the Discussion section but not described in the Results nor in any table. On line 218: where are the alcohol results coming from?

Paragraph one lines 232-240 repeats mostly the one on line 172-177. 

Author Response

Point by point response:

It is an interesting study on changes in lifestyle and anthropometric measures between before and during lockdown in Italy. The study relies on a self-reported retrospective questionnaire, subject to a large amount of bias.  

R. The title does not reflect this and should be self-explanatory. 

A. The Authors agree with this comment and changed the title according to the reviewer’s suggestion. The new title is: “Lifestyle changes and Body Mass Index during Covid-19 pandemic lockdown : an Italian online-survey”.

R. Language needs revision, it is inaccurate in many instances, and lacks continuity, sometimes lacks of verbs. Inappropriate use of connectors such as Nevertheless. Example of issues (but the entire manuscript needs extensive revision). 

A. The test was revised by an English native-language speaker.

R. Intro and abstract: Some countries in the world did not apply strict lockdowns, I would talk about "contingency measures including total or partial lockdowns". 

A. The authors agree with the reviewer’s comment and replaced “lockdown” with “contingency measures including total or partial lockdowns", as suggested (page 1, lines 33-34).

R. Line 37: what is "smart working"?

A. The “Smart-working” term is used in Italy for “working from home”. The authors thank the reviewer for this linguistic mistake and corrected the term throughout the whole manuscript. 

R. Line 38: "studies are raising" --> emerging

A. The authors replaced the “raising” with “emerging” (page 1, line 39).

R. One-sentence paragraphs should be avoided. The introduction is made up of 4 one-sentence paragraph which does not link well with each other. 

A. Thank you for your comment; we tried to eliminate one-sentence paragraphs from all the manuscript and revised the introduction section.

R. Line 43: Why is this sentence at present tense? Are they the results from a study, from which country/setting?

A. The authors agree with the reviewer's comment and have completely revised the Introduction section.

R. Line 111: Multiple regression should be multivariable? All lifestyle factors included in the same model? Or different regression models separately?

A.Multiple and multivariable are synonyms and refer to a model in which multiple variables are found on the right side of the model equation and one variable is on the left side. We now specified that more independent variables were simultaneously evaluated in a regression model (page 3, lines 118-119; page 3, line 141).

R. Some information is missing on the study population. Is it intended to be representative of the population of specific region, all of Italy...? Very little control is possible to check for accuracy of age, sex, place of residence, employment, as participants in the Google forms are anonymous and can easily report false information. 

A. The authors agree with the reviewer’s comment and better clarified these aspects as limitations of the study (page 7, lines 241-243). 

R. How were "potential duplicates" identified? (line 116)

A. Potential duplicates were identified comparing timestamps, item response patterns and participants e-mail addresses. We clarified it in the manuscript (page 3, line 123).

R. Table 1. It is hard to understand what is described here. The Negative change / no change / positive change is for each individual lifestyle habit? So that does not correspond to a total score <10, 10-11, >=12. Rather it should be individual score of 0, 1, or 2. 

A. For lifestyle habits and eating behaviours changes, data are described as absolute number (n) and relative frequency (%) of subjects who changed the lifestyle habits listed in the first column. The authors better clarified this aspect in table 1.

R. The results of the "multiple regression" are not presented in any table or figure, just some of them are mentioned in the text. These data need to be presented, and the models clarified.

A. The authors agree with the reviewer’s comment and added table 2 with the multiple regression analysis (page 5). 

R. Also, some more details are needed on the time frame. T0 "pre-lockdown" is quite a vague date. What was the exact question? Over the month of January/February 2020? 

A. The authors thank the reviewer for the comment and better clarified the time-frame of T0 and T1 throughout the manuscript. In brief, T0 was referred to lifestyle habits/eating behaviours before the start of “Phase 1” of the Covid-19 pandemic (8 March 2020) while T1 to lifestyle habits/eating behaviours during “Phase 1” of the Covid-19 pandemic (8 March - 4 May 2020) (abstract section and page 2, lines 71-71).

R. Line 165: what do you mean by "as previously reported"? In another study? Reference?

A. The authors have rephrased the sentence correctly (Page 6, lines 187-193).

R. Line 168: "BMI was significantly negatively associated" is wrong wording. It should be something like "People who worsened some of their behaviours, namely water consumption, alcohol etc, displayed an increase in BMI compared to pre-lockdown".

A. The authors thank and agree with the reviewer comment and rephrased the sentence correctly (Page 6, line 193-196).

R. Some ideas are good in the discussion but the formulation is shaky. E.g. line 200 trying new food and recipes is not a "downside". 

A.The authors appreciate the reviewer's comment and revised the Discussion section.

R. Why are some results discussed in the Discussion section but not described in the Results nor in any table. On line 218: where are the alcohol results coming from?

A. The discussion section has been revised.

R. The sentence line 207 to 212 is very long, lacks a verb and does not make a lot of sense.

A. The sentence has been reworded.

R. Paragraph one lines 232-240 repeats mostly the one on line 172-177. 

A. This paragraph has been reworded.

The authors thank the reviewer for his/her comments and reviewed the whole manuscript, as suggested. 

Reviewer 2 Report

Thank you for the opportunity to review this manuscript. The topic is important; however, the authors may need clarify some issues.

My major concerns:

  1. Age is an important determinant for lifestyle and diet behaviours as well as BMI. However, the authors did not adjust for this important factor in the analysis. Was information on age collected in the study?

  1. There are too many paragraphs in the Introduction section, the authors may consider restructuring the section.

  1. Self-reported data on lifestyle habits, diet behaviours, and BMI before and after lockdown were all collected after lockdown, which may result in measurement biases. How did the authors address this issue?

  1. The authors may need to report the proportion of missing values for variables listed in Table 1. How did the authors address the missing values?

Minor concerns:

  1. The authors stated “The survey was launched on 30th April 2020”, so when did the survey was completed? Individuals who filled out questionnaires at a time point far from the end of the lockdown might be more likely to report biased information.

  1. In Table 1, the subtitle “BMI changes” seems not to be appropriate as BMIs both before and after lockdown are presented.

  1. In Table 1, Median (IQR) for “Lifestyle habits and eating behaviours score” may not be correct. Please check.

  1. There are many errors in Table 1. For example, the sum of the numbers of participants for the “daily breakfast consumption” row is 1306 and that for the “Habitual vegetables consumption at lunch” row is 1241. This is inconsistent with the total number of participants included in the analysis (1304).

Author Response

REVIEWER #2 - Thank you for the opportunity to review this manuscript. The topic is important; however, the authors may need clarify some issues. 

MAJOR CONCERNS.

R. Age is an important determinant for lifestyle and diet behaviours as well as BMI. However, the authors did not adjust for this important factor in the analysis. Was information on age collected in the study?

A.  The authors agree with the reviewer comment. However, as stated in the methods section, they surveyed the adult population (aged>18 years) (page 2, line 74) with no age collection. For this reason, the analysis was not age-adjusted; this limitation has been included in the discussion section (page 7, lines 244-245).  

R. There are too many paragraphs in the Introduction section, the authors may consider restructuring the section.

A. The authors agree with the reviewer’s comment and edited the introduction section as a whole.  

R. Self-reported data on lifestyle habits, diet behaviours, and BMI before and after lockdown were all collected after lockdown, which may result in measurement biases. How did the authors address this issue?

A. The authors agree, but what is usually captured by retrospective surveys is the perception of the subject. Anyway, this has been reported by the authors in the limitation section and discussed at the end of the Discussion section.   

R. The authors may need to report the proportion of missing values for variables listed in Table 1. How did the authors address the missing values?

A. Among lifestyle habits and eating behaviours variables, the only one with missing item responses was item #9 (Habitual vegetable consumption at lunch), with a total of 1241/1304 (4.8%) and, as suggested by the reviewer we specified it in the manuscript (page 3, line 126-127). 

MINOR CONCERNS. 

R. The authors stated “The survey was launched on 30th April 2020”, so when did the survey was completed? Individuals who filled out questionnaires at a time point far from the end of the lockdown might be more likely to report biased information.  

A. The authors added the period of data collection ranging from 30 April 2020 to 10 May 2020 (page 2, line 75)

R. In Table 1, the subtitle “BMI changes” seems not to be appropriate as BMIs both before and after lockdown are presented.

A. The authors agree with the reviewer comment and edited Table 1 subtitle.

R. In Table 1, Median (IQR) for “Lifestyle habits and eating behaviours score” may not be correct. Please check.

A. The authors thank the reviewer for the comment, we checked the IQR, but since the score is a discrete variable, it ranges between 4 and 16, the third quartile is 11 and the first quartile is 9, IQR is 2.  

R. There are many errors in Table 1. For example, the sum of the numbers of participants for the “daily breakfast consumption” row is 1306 and that for the “Habitual vegetables consumption at lunch” row is 1241. This is inconsistent with the total number of participants included in the analysis (1304).

A. The authors thank the reviewer for the comment and corrected typos in table 1 (page 5, line 8). As to “daily breakfast consumption” there was a typo, while as to “Habitual vegetable consumption at lunch”, missing data are responsible for the lesser total sum of 1241. 

Round 2

Reviewer 2 Report

I thank the authors for their responses to my comments. I had just one additional point, reviewing their revisions.

The authors conducted multiple regression analysis (Table 2) such that they may need to control false discovery rate.

Author Response

Reviewer 2 R. "The authors conducted multiple regression analysis (Table 2) such that they may need to control false discovery rate".

A. We coducted a multiple regression analysis, meaning that we run one single regression model with one dependent variable and variuos independent variables (in Methods section, Statistical Analysis, we clarified that – “independent variables simultaneously put in the mode”. In this case there is no need to control for False Discovery Rate, furthermore this is an explorative study and not a confimatory one. The author has also slighltly changed the title of table 2 (page 5, line 163).